# Automatic Image-Based Event Detection for Large-N Seismic Arrays Using a Convolutional Neural Network

**Miłosz Mężyk *** , **Michał Chamarczuk and Michał Malinowski**

Institute of Geophysics PAS, Ksiecia Janusza 64, 01-452 Warsaw, Poland; mchamarczuk@igf.edu.pl (M.C.); michalm@igf.edu.pl (M.M.)

* Correspondence: mmezyk@igf.edu.pl

**Abstract:** Passive seismic experiments have been proposed as a cost-effective and non-invasive alternative to controlled-source seismology, allowing body–wave reflections based on seismic interferometry principles to be retrieved. However, from the huge volume of the recorded ambient noise, only selected time periods (noise panels) are contributing constructively to the retrieval of reflections. We address the issue of automatic scanning of ambient noise data recorded by a large-N array in search of body–wave energy (body–wave events) utilizing a convolutional neural network (CNN). It consists of computing first both amplitude and frequency attribute values at each receiver station for all divided portions of the recorded signal (noise panels). The created 2-D attribute maps are then converted to images and used to extract spatial and temporal patterns associated with the body–wave energy present in the data to build binary CNN-based classifiers. The ensemble of two multi-headed CNN models trained separately on the frequency and amplitude attribute maps demonstrates better generalization ability than each of its participating networks. We also compare the prediction performance of our deep learning (DL) framework with a conventional machine learning (ML) algorithm called XGBoost. The DL-based solution applied to 240 h of ambient seismic noise data recorded by the Kylylahti array in Finland demonstrates high detection accuracy and the superiority over the ML-based one. The ensemble of CNN-based models managed to find almost three times more verified body–wave events in the full unlabelled dataset than it was provided at the training stage. Moreover, the high-level abstraction features extracted at the deeper convolution layers can be used to perform unsupervised clustering of the classified panels with respect to their visual characteristics.

**Keywords:** interferometry; ambient noise; convolutional neural network; deep learning; detection; data selection; body–wave; passive seismic monitoring

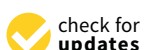

## 1. Introduction

The demand for cost-effective exploration technologies that could characterize geological structures with minimum impact on the environment has drawn considerable attention to seismic imaging methods employing uncontrolled sources. It is possible to mimic, to some extent, controlled-source seismic data at the site where a passive seismic experiment is carried out by applying seismic interferometry (SI) to ambient noise recordings [1]. As long as the wavefield is diffuse, ambient noise SI (ANSI) allows the impulse response of a medium to be extracted from the noise recorded at two receivers based on their averaged cross-correlation as if one of them was a source [2–4]. SI by cross-correlation, as an alternative to active-source reflection seismic surveys, has been successfully applied in numerous studies ranging from subsurface imaging [5–12] for monitoring purposes [13,14]. If the required condition of diffuse wavefield is met, the recovered response is regarded to represent empirical Green's function (EGF) accounting for both surface and body wave propagation in the medium between two receivers [15]. The EGF can be retrieved either from multiply scattered waves by stacking all short-duration cross-correlation functions

(CCFs) over a sufficiently long period of time [15] or from waves produced by some specific uncontrolled sources [16]. The latter approach offers a solution without the need to conduct long-term data acquisition. The main idea is the selective stacking of CCFs that are coherent to each other, for instance, only those related to body or surface waves, and rejecting the time segments dominated by sources located outside the stationary-phase regions [17].

Concurrent development of acquisition techniques, such as a new generation of accelerometers based on a micro-electromechanical system (MEMS) [18] or fiber optic-distributed acoustic sensing (DAS) [19], has lowered the cost of low-power sensors and made it more feasible to deploy arrays that are dense enough to collect high-frequency seismic data free from spatial aliasing. However, as the number of simultaneously recording sensors (channels) is growing, the volume of data starts to surpass the processing capabilities of techniques that require human interaction [20]. The rapid increase in the number of large seismic datasets [21] requires new strategies and methods that would make it feasible to automate the execution of processing workflows and reduce the turnaround time. In the case of ambient noise recordings focused on body–wave reflection retrieval, the selective-stacking approach can be automated and improved by introducing the algorithms that could sift through massive volumes of continuous data in order to identify coherent noise sources, which contribute constructively to the stacked EGFs. The automatic detection of noise records (noise panels) containing body–wave energy (hereinafter referred to as body–wave events) evaluated using large-N (i.e., large receiver number) array was already tackled by Chamarczuk et al. [22], who applied support vector machine (SVM), a supervised machine-learning (ML) algorithm, in combination with a cross-correlated wavefield evaluation in the tau-p (intercept time—slowness) domain and predefined body–wave velocity limits. Recently, deep learning approaches have been also utilized for detection purposes in passive seismic recordings. However, they are restricted only to the extraction of surface waves from DAS data [23–25] or earthquake-induced seismicity [26–35]. The other demonstrated solutions for data selection are mostly semi-automatic and/or performed on preprocessed and cross-correlated data, which in turn requires extra operator workload and computational cost. Examples of these methods are based on instantaneous phase coherence [36], statistical time-series classification [37], illumination diagnosis [38], asymmetry of CCFs [39], local similarity function [40], or on signal-to-noise ratio (SNR) criterion [41–44].

In this study, we employ the convolutional neural network (CNN) with the specific Visual Geometry Group (VGG) building blocks introduced in [45] to develop a predictive model for the near real-time body–wave event detection in passive seismic data. We exploit the fact that coherent signals have certain identifiable temporal and spatial characteristics as the signals move across the array. These characteristics can be well represented by the basic frequency and amplitude attribute maps computed sequentially for the whole array in consecutive short time segments (noise panels) and used as feature providers for training and prediction purposes with a CNN algorithm. Once the model is learned, it allows for the fast and robust binary classification of noise panels on the basis of their body and surface energy image content prior to the cross-correlation operation. We also show that the deep features of the input images extracted by keeping the activation values of the last pooling layer might be utilized for grouping the identified body–wave events according to the similarity of their visual characteristics. The verified application of our approach is limited to the high-density seismic surveys, but if that is not the case, the proposed model creation scheme can be still potentially adapted in other settings. What is more, we provide our codes publicly via the Jupyter notebooks and the GitHub platform to facilitate the reproduction of our results (see link in Supplementary Materials).

The paper is organized as follows. First, we introduce the dataset and signal processing techniques for deriving the attribute maps. Then, we describe the model building process with special emphasis on the implementation of VGG architecture. Subsequently, we apply the proposed CNN-based models to the field data recordings and confront its performance with the gradient-boosting classifiers trained on the flattened attribute maps.

Finally, we demonstrate unsupervised clustering of neuron activations and conclude on the effectiveness of the proposed CNN-based approach in terms of automatic event detection.

## 2. Dataset and Attribute Maps

The Kylylahti large-N array (see Figure 1a for the layout) was deployed in the vicinity of the polymetallic underground Kylylahti mine in Polvijärvi (Eastern Finland) as a part of the COGITO-MIN project. Its primary purpose was to advance the development of ANSI imaging techniques for mineral exploration and provide a baseline for testing novel array-processing techniques. The Kylylahti array was formed by 994 receiver stations distributed regularly over a 3.5 × 3 km area with 200 m line spacing and 50 m receiver interval. Surface conditions varied from exposed bedrock to swamps. Each receiver station consisted of a Geospace GSR recorder and 6 × 10-Hz geophones bunched together and buried whenever possible, and it was recording at a 2-millisecond sample rate for about 20 h/day. In this work, we analyze the dataset containing 240-h long time series resampled to 4 ms, which sums to almost 214 billion samples for all available stations. The time series acquired by the sensors are arranged in a two-dimensional data array, with a dimension of time (samples) and a dimension of space (channels). The measurement channels are numbered from 1 to 994 in a line-wise manner starting with the first seismic sensor unit in the profile L01 and ending with the last seismic sensor in the profile L19. Besides the advantage of high-resolution imaging obtained by the measurements that are densely spaced both in the inline and crossline directions, like in the case of the Kylylahti array, these datasets can be also potentially treated as images for deep learning purposes if they are appropriately represented.

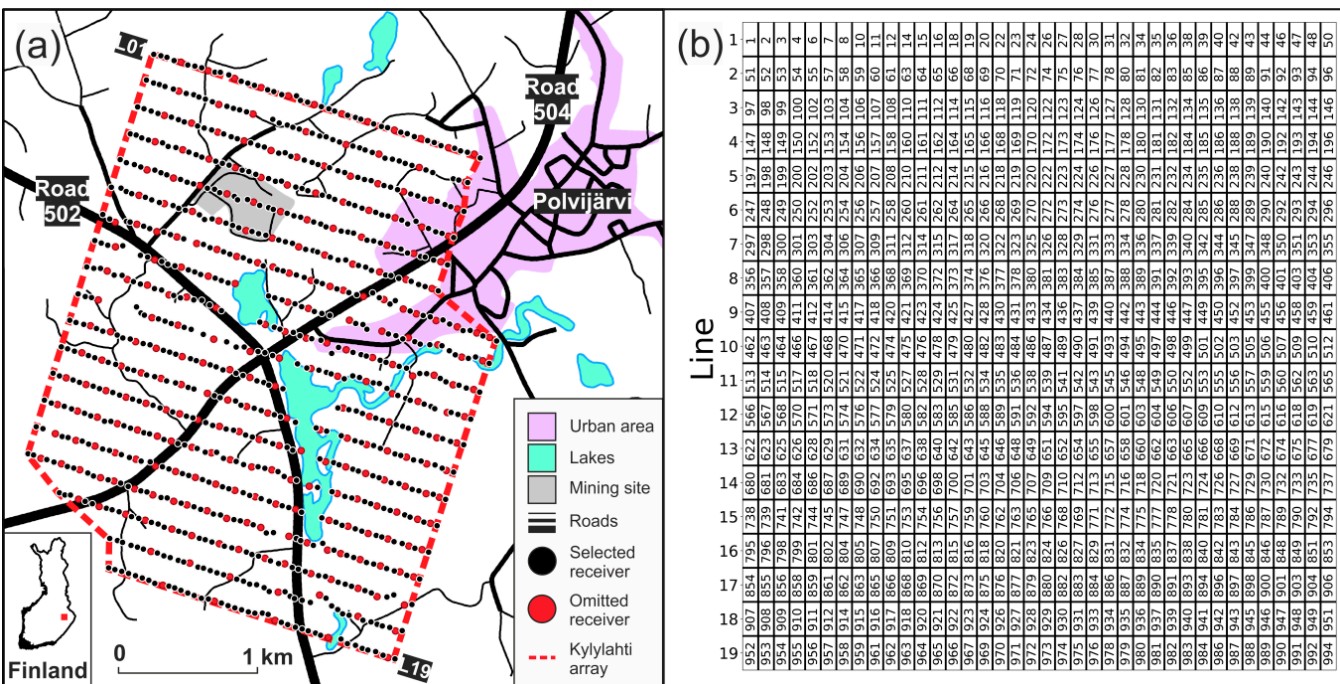

**Figure 1.** (**a**) Map of the study region. Red and black dots are the receivers forming the Kylylahti array. To maintain an equal number of receiver stations per seismic line, some of the receivers (depicted in red) are skipped and do not contribute to the computation of attribute maps. The red dashed line represents the contour enclosing all the receiver lines of the array (labeled as L01–L19). The inset shows an outline map of Finland and the location of the plotted region (red square). (**b**) Matrix representation of the subsampled Kylylahti array. Each of the selected receiver stations is mapped to the corresponding cell number (hereafter referred to as a receiver identification number). The rectangular shape of cell tiles is caused by duplication of each row (receiver line). It is needed to match the number of columns (receivers) and preserve the aspect ratio of a square.

To tackle the task of automatic pattern recognition in seismic signals using a CNN algorithm, it is required to transform the analyzed signals from their non-image form to image frame type data. Although the most straightforward way of achieving that is the direct plot in the time–space domain of the noise recorded along the seismic lines, it might be quite cumbersome to accomplish, especially for passive seismic experiments due to a relatively small number of receiver stations per line, as compared to active surveys, and the implied lack of controlled sources illuminating the subsurface or interference factors. In many cases, the image patterns evaluated along the single receiver line might be not evident or ambiguous due to their complexity. Thereby, the training dataset constructed from the straightforward visual representations of combined uncorrelated ambient noise measurements could lead to biased prediction models that do not perform well in practice. Here, we address the issue of generalizability by combining the individual seismic trace characteristics into the map-like representation of noise panels. The resulting maps not only reduce the redundant and incoherent information contained in passive seismic data but also provide a comprehensive approach for the simultaneous analysis of signals across a dense seismic array. First, we segment a continuous recording at a receiver station into separate intervals (here we use 10-s long ones), and apply signal transformations (Figure 2) to derive their single value characteristics suitable for populating the two-dimensional data structure called an attribute matrix. This procedure is repeated for all selected receivers in the seismic array. The selection is necessary in order to match the number of attribute values (matrix columns) to the number of seismic lines (matrix rows) or multiples of their quantity. In such a way we get the square matrix that can be directly saved as an image, referred to as an attribute map. In the case of the Kylylahti array, we select 722 receivers (out of 994) (Figure 1a) for the calculation of attribute values representing frequency and amplitude information extracted from segmented seismic signals (Figure 2). To reveal the frequency characteristics, we compute the amplitude spectra of these fragmented signals and then sum the resulting amplitudes in the following frequency ranges: 10–20, 20–30, and 30–40 Hz (Figure 2d,g). The inspection of manually detected body–wave events showed that their bandwidth overlaps mostly with these defined frequency bands. We also tested narrower intervals, like, for instance, bins spaced every 5 Hz from 10 to 50 Hz, but the corresponding frequency maps turned out to be too similar to each other to deliver a meaningful improvement in detection capabilities over the prediction model utilizing only three 10-Hz-wide intervals.

Prior to the extraction of the remaining attributes used in this study, the input signals are normalized by the root-mean-square (RMS) value to equalize the amplitude. It allows for the comparison of signals recorded by various receiver stations at different times during the measurement period. Analogously to the frequency attributes, we derive three amplitude characteristics from each RMS-scaled signal. The first one is the maximum absolute value (Figure 2b, a) and two others determine the number of samples of the signal envelope that fall in the percentile range 0–33th and 33–66th (Figure 2c,f). The envelope amplitudes above the percentile 66th correspond mostly to the signals without coherent energy, and thus this range can be skipped. We employ the envelope in our analysis to better discriminate between the coherent and the incoherent signals depicted as gray and black lines in Figure 2c, respectively.

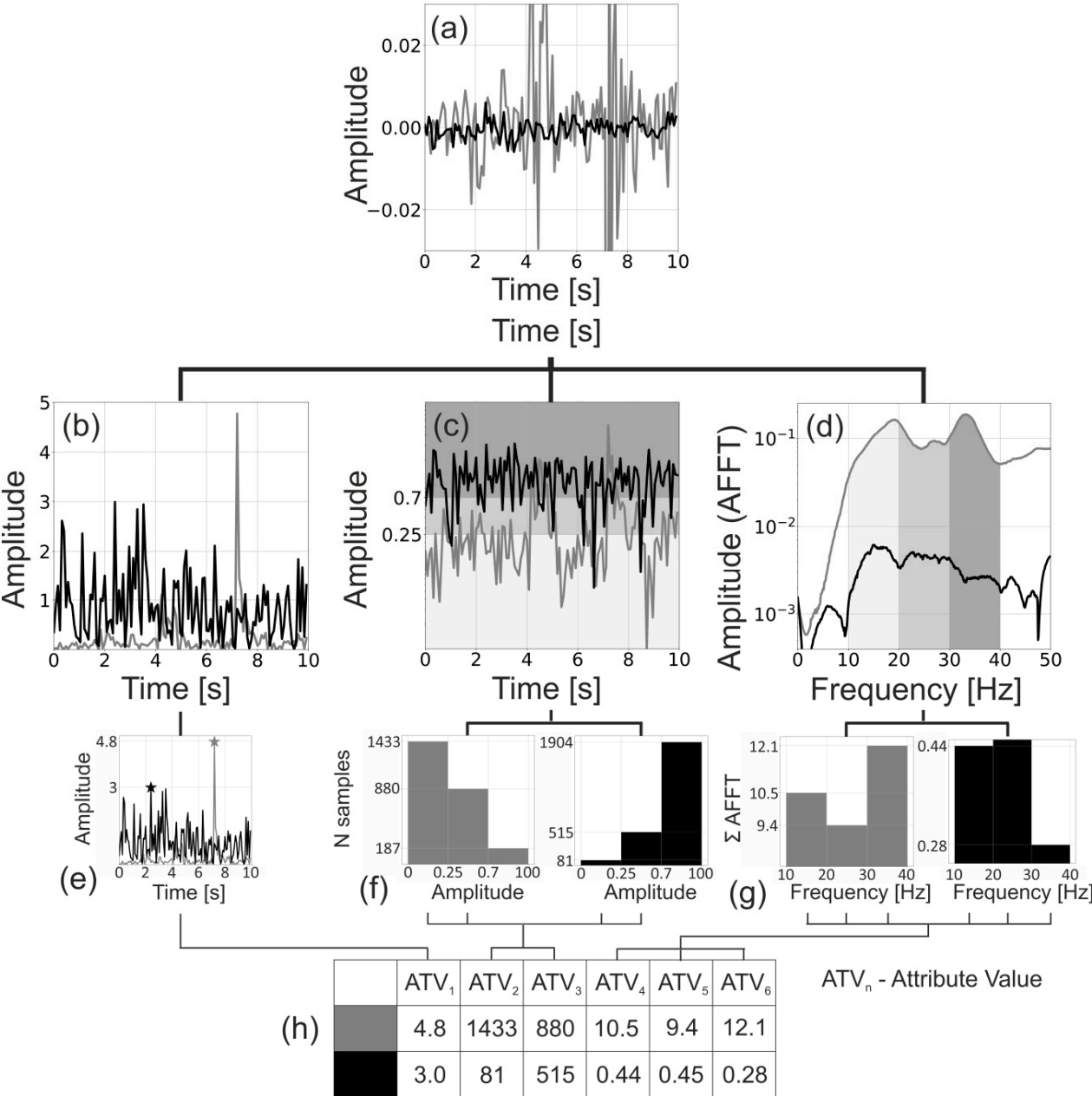

**Figure 2.** Computation of attribute values for a single trace in a noise panel (defined as recording segment of 10 s for a single receiver station). (**a**) Unscaled input signals. The gray and black curve represents the signal with identified coherent wave arrivals and random background noise, respectively. The following subfigures are displayed using the same colour convention. (**b**) The absolute values of the RMS-scaled signals of (**a**). (**c**) The envelope values of the RMS-scaled signals of (**a**). The colour shading corresponds to the three amplitude ranges. (**d**) The amplitude spectra. The colour shading corresponds to the three frequency ranges. (**e**) The maximum values of (**b**) are shown as star markers. (**f**) The histograms of samples in (**c**) that fall in each specified amplitude range. (**g**) The bar plots describe the amplitude spectra of (**d**) summed in each specified frequency range. (**h**) The feature matrix, derived from (**e**–**g**).

Once a single noise panel is characterized by the attributes (Figure 2h) computed for all selected receiver recordings, they are mapped to the corresponding cell number in the attribute matrices and plotted with a chosen colour palette. The resulting attribute maps are then used to form the training dataset and to serve as an input to the final prediction model. As shown in Figure 3, the attribute maps generated for the ten hand-picked body–wave and non-body–wave panels provide a robust spatio-temporal descriptor of the multi-scattered energy in passive seismic data.

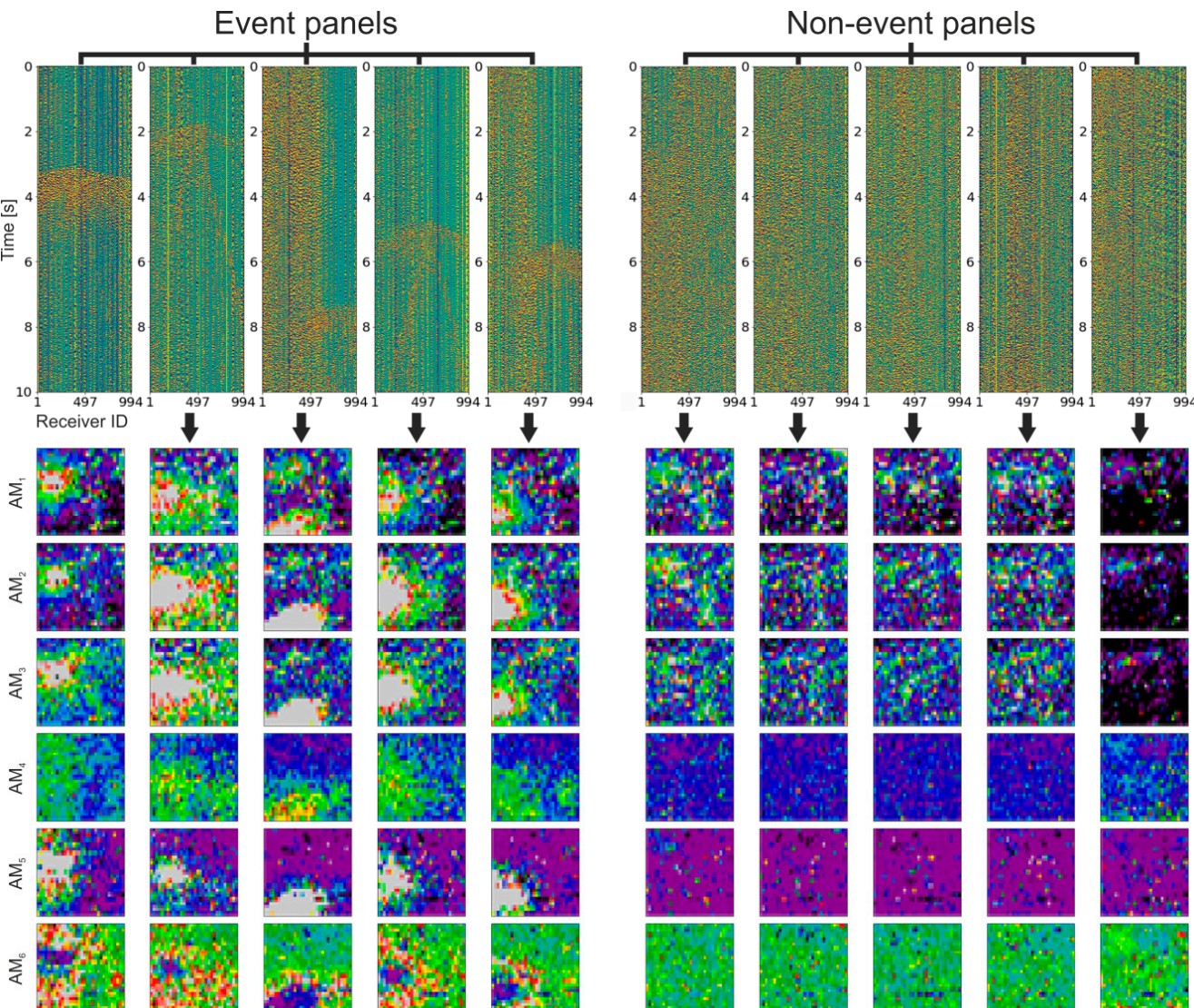

**Figure 3.** Transformation of the arbitrarily selected noise panels with/without body–wave energy into their image representations achieved through the defined set of attribute maps (AMs). The AM subscript corresponds to the same order in which derivation of the attribute values (ATVs) is shown in Figure 2. The attribute maps on the left are the examples of the body–wave image class in the labeled training set while the attribute maps on the right demonstrate the "noise" image class.

## 3. Deep Supervised Learning

In this section, we present and describe both the training data set and the employed CNN architecture for the body–wave event detection in the obtained image representations of ambient noise recordings. We also discuss the model building process.

### 3.1. Training Dataset

To build a prediction model, a set of 1900 training examples is manually selected out of 86,000 non-overlapping noise panels that are randomly evaluated. 14% of the selected panels are labeled as the body–wave while the rest is classified as the background noise. The disproportionate ratio of observations in each class, called the class imbalance, might have a significant impact on a classifier by affecting both its convergence during the training phase and generalization of a model on the unseen data [46]. To handle CNN training on the imbalanced dataset, we choose the basic version of the oversampling method which

simply duplicates samples from the minority class [47]. Here, we replicate the body–wave class seven times to match the size of the noise class. Subsequently, each of the selected panels is transformed into the six attribute maps, described in the previous section, and saved as thumbnail red-green-black (RGB) images at the spatial resolution of $32 \times 32$ pixels to form the final training set comprised of almost 20,000 images. In Figure 3, we show some examples from both classes.

### 3.2. CNN Architecture

A Visual Geometry Group (VGG) net is a CNN architecture that was proposed in [47] and designed for large-scale image classification. It was coincident with the idea of using blocks (i.e., repeated structures in the network). The authors demonstrated that their models can be applied to a wide range of tasks and datasets with a performance comparable to the most efficient architectures in computer vision such as GoogleNet [48] or, not existing at that time, ResNet [49]. In this study, we develop a new model from scratch that is based on the VGG concept. We consider it as a sensible starting point due to its straightforward architecture that can be easily implemented and modified to optimize its usefulness for our defined detection problem. Variations of the VGG network architecture can be obtained by differences in the number of convolutional layers and output channels in each block and also by the number of implemented blocks itself. Since we train our model on the low-resolution images that are seven times smaller than the images in the ImageNet challenge dataset [50] used for the state-of-the-art performance test of the VGG models, we decide to substantially reduce the complexity of the network. The reduction was also needed to conduct series of computationally intensive prototyping experiments on a multi-core central processing unit (CPU) cluster in an acceptable time frame. As shown in Figure 4, the proposed CNN architecture contains an input layer of size $32 \times 32$ receiving a 3-channel (RGB) image (attribute map) and hidden layers grouped into three VGG blocks each defined as a group of convolutional layers that use filters with a small receptive field of size $3 \times 3$ followed by a max-pooling (subsampling) layer. The pooling is performed over a $2 \times 2$-pixel window with a stride 2 [45] in order to reduce the dimensionality of the input representation (e.g., image or hidden-layer output matrix) without affecting the important features [51]. The three VGG blocks have convolutional layers with 64, 128, and 256 channels (filters), respectively, and each of the blocks is also accompanied by the activation function (here we use a Rectified Linear Unit called ReLU) and the dropout regularization that randomly sets input neurons to 0 with a specified rate (here set to 20%) at each step during training time, which helps prevent overfitting [52]. The increasing depth of the layers is common for CNN architectures as it influences the power of abstraction and enables the extraction of a wider diversity of features in images. After flattening the output of the last VGG block, the classification network was constructed using two consecutive dense layers. The last fully connected linear layer uses the softmax activation function, designed for the multi-class classification cases, that returns a score level vector with its sum normalized to 1. These values can be interpreted as class probabilities and used for the final classification based on predetermined thresholds (e.g., a default 0.5 value for two-class problems) or the maximum probability criterion (e.g., for multi-class problems).

### 3.3. Building a Prediction Pipeline

To improve the training procedure, we propose the ensemble of two multi-headed CNN models trained separately on the amplitude- and frequency-based attribute maps (Figure 4, $AM_1$–$AM_3$ and $AM_4$–$AM_6$, respectively). Each of them is created by unifying three independent sub-models (so-called heads) into the jointly trained framework, where they cooperate to improve the learning outcomes. The input heads of sub-models can be considered as the net branches using the same types of layers but with different weights and initialization. The single sub-model reads the individual input sequence (i.e., a certain sub-category of attribute maps) and returns the flat vector containing a summary of the learned features from the sequence (Figure 4, $AV_1$–$AV_6$). These internal representations

are combined after being interpreted by the first fully connected layer (Figure 4, $FCL_1$) and used to make a prediction in the softmax classification layer (Figure 4, $FCL_2$). Both classifiers' predictions (Figure 4, $p_A$ and $p_F$) are then again combined by averaging to output the final prediction estimate (Figure 4, $p_{avg}$).

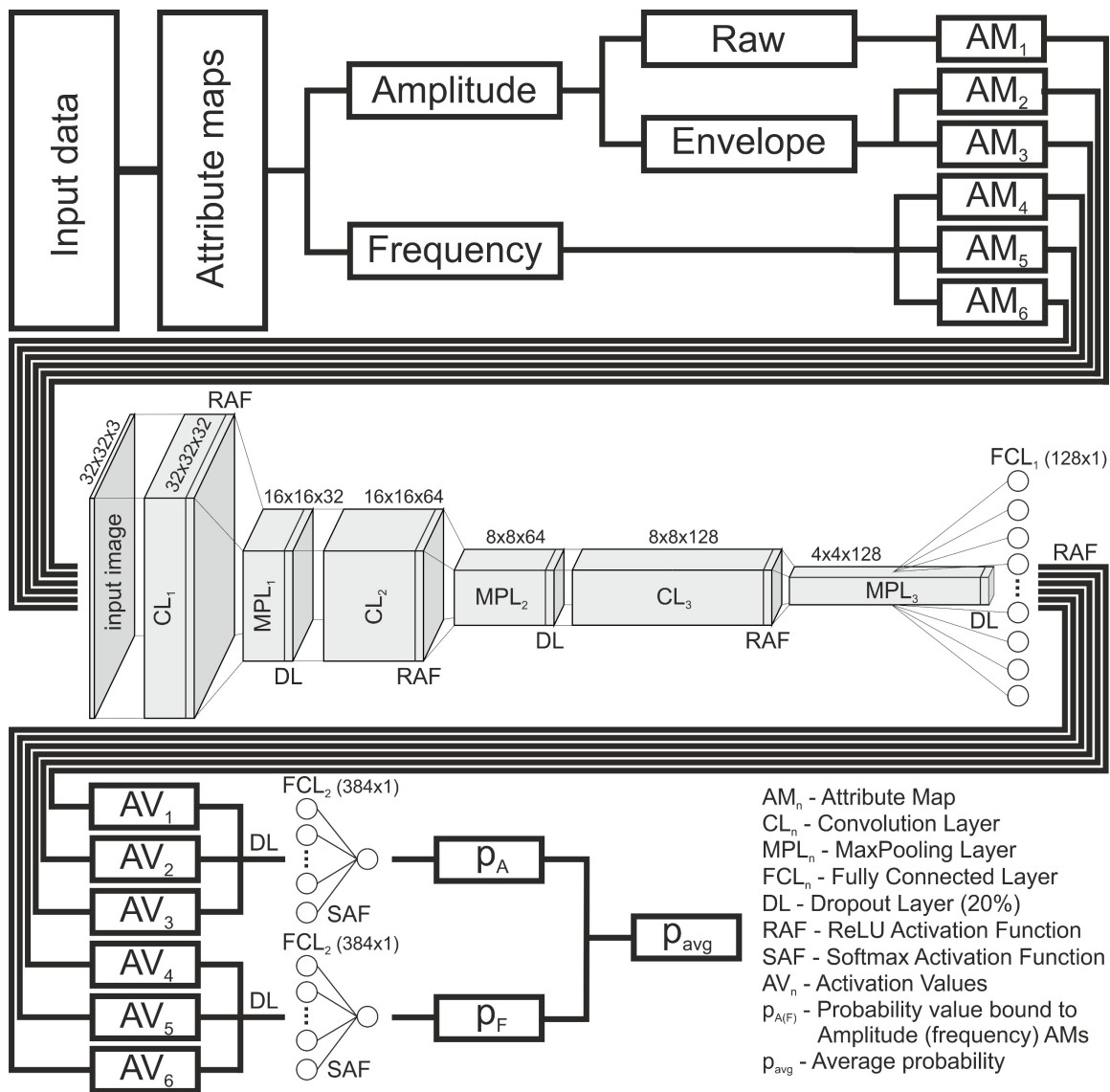

**Figure 4.** Flow diagram illustrating a deep learning-based solution model and the employed Visual Geometry Group (VGG)-16 convolutional neural network (CNN) architecture.

By using multiple heads, we allow each sub-model to extract various high-dimensional spatial features within the two main categories of attribute maps (i.e., amplitude- and frequency-derived ones). The ensemble averaging model integrating the strength of the two constituent models, when subjected to the images unseen during the training stage, generates probability estimates that are more accurate than those produced by the individual models (see Section 4.2). Moreover, it can complete the prediction in a shorter time due to the reduction in the number of predictive models (the frequency and amplitude-based multi-headed models versus the mixture of the six single-headed individual models). Figure 4 summarizes the prediction pipeline that was built in Python3 using the Keras neural networks library [53] running on the top of the TensorFlow [54] library as its interface. The parameter configuration of the proposed CNN-based models is listed in Table 1. These parameter values were iteratively adjusted to improve the model's generalizability.

**Table 1.** Parameters of the proposed CNN-based models.

| Parameter | Value/Type |
|---|---|
| Input image size | $32 \times 32$ px $\times$ 3 channels (RGB) |
| Number of epochs | 200 |
| Batch size | 16 |
| Kernel initializer | He uniform |
| Activation function | ReLU |
| Classification function | Softmax |
| Loss function | Categorical cross-entropy |
| Dropout rate | 0.2 |
| Optimizer | SGD |
| Learning rate | 0.001 |
| Momentum | 0.9 |

## 4. Results

In the following subsections, we describe the metrics used for evaluating and comparing both the CNN- and ML-based models. We also provide the summary statistics of the overall prediction results achieved with the ensemble of CNN-based models and present the outcome of the unsupervised deep embedded clustering that aims to categorize the detected noise panels.

### 4.1. Metrics

In a binary classification problem, any prediction relative to labeled data can be either positive or negative. The predictive accuracy of a ML algorithm is usually assessed based on the difference between the observed (actual) and predicted values. Given a set of instances (i.e., noise panels to be classified), a two-by-two confusion matrix (also called a contingency table) can be constructed to represent the decisions made by the classifier. The structure of this matrix is shown in Table 2 and it serves as a basis for calculating several accuracy metrics.

**Table 2.** Confusion matrix structure for binary classification problems.

| | Predicted Class | |
|---|---|---|
| **Actual Class** | True Negative (TN) | False Positive (FP) |
| | False Negative (FN) | True Positive (TP) |

To measure the overall diagnostic performance of a predictive classifier we consider two commonly used methods: the $F_1$-Score [55] and the area under the Precision-Recall curve (AUC-PR). The latter has been reported as an alternative to Receiver Operator Characteristic (ROC) curve for imbalanced classification with a severe skew in the class distribution [56] where the positives (i.e., body–wave panels) are more relevant than the negatives. The former belongs to the single threshold category of metrics and it is defined according to the formula (1) as the weighted harmonic mean of precision (2) and recall (3). Typically $\alpha$ takes the value 1 to signify the balance between precision and recall. Precision expresses the proportion of correctly classified instances that are relevant (called true positives and defined here as recognized body–wave panels) to the sum of true positives and false positives (instances misclassified as relevant), while recall is the ratio of true positives to the sum of true positives and false negatives (instances misclassified as non-relevant). False positives and false negatives are represented in our case by non-body–wave panels recognized as body–wave panels and body–wave panels recognized as non-body–wave panels, respectively.

$$F_\alpha = \frac{(1+\alpha)(precision \cdot recall)}{(\alpha \cdot precision) + recall} \tag{1}$$

$$precision= \frac{TP}{TP+FP} \tag{2}$$

$$recall= \frac{TP}{TP+FN} \tag{3}$$

The $F_1$-Score and the similar metrics are based on a single threshold level (usually equal to 0.5), where instances above the threshold are predicted as positive (relevant) and the rest as negative (non-relevant). Unlike these metrics, the AUC-PR provides a score interpreted as an overall model performance across all possible thresholds. While the PR curve depicts precision as the function of recall, the AUC can be interpreted as the average value of precision over the interval from recall = 0 to recall = 1.

### 4.2. Prediction

We estimated the performance of the proposed model first on the training data set by employing k-fold cross-validation (CV) implemented in the scikit-learn library [57]. It is a common technique that randomly divides the input data set into k non-overlapping equal subsets (here we select the value of k equal to 10) in which one part serves as an independent test set that is held-back for evaluation, whereas the remaining folds are used collectively for training a model. This process is repeated k times, i.e., until the model is not trained and tested on all subsets. Then the average of the prediction scores, recorded in every run, is calculated and treated as the reliable performance score. It should be noted here that while the models are trained on the class-balanced training sets, the validation is performed on the class-unbalanced (unseen) sets for the sake of clarity. Table 3 lists the average scores that are derived across folds for both the F1 and AUC-PR metrics. In the same table, we also report the scores for the ensemble of models trained with the extreme gradient boosting algorithm (XGBoost) [58] on the same set of noise panels as the CNN-based counterpart but represented in another form. Instead of images, the XGBoost reads a set of feature vectors, which in our case are flattened attribute matrices, like the one shown in Figure 1b. We choose this specific ML algorithm for comparison purposes because of its broad versatility in the modeling of nonlinear processes and overall high prediction efficiency reported in many recent seismic research studies [59–63]. Additionally, confusion matrices are summed over all cross-validation folds to get the overall (cumulative) confusion matrix presented in Table 4. Both the F1 scores and the confusion matrices were determined based on the classification threshold set to 0.5.

**Table 3.** Comparison of the prediction performance between the CNN- and XGBoost-based models when applied to the same unseen data. To evaluate the accuracy of the classifiers we use 10-fold cross-validation as well as F1 and AUC-PR metric scores that are recommended for the imbalanced data.

|  | Amplitude-Based CNN | Frequency-Based CNN | Ensemble-Based CNN | Ensemble-Based XGBoost |
|---|---|---|---|---|
| $F_1$ | 0.890 | 0.894 | 0.963 | 0.849 |
| AUC-PR | 0.961 | 0.964 | 0.997 | 0.939 |

**Table 4.** Cumulative confusion matrices calculated during the cross-validation for the CNN- and XGBoost-based models.

| Ensemble-Based CNN | | Ensemble-Based XGBoost | |
|---|---|---|---|
| TN = 1653 | FP = 16 | TN = 1657 | FP = 12 |
| FN = 2 | TP = 229 | FN = 51 | TP = 180 |

In the final prediction and evaluation stage, we employed the obtained ensemble of amplitude- and frequency-based CNN models to classify the entire data set comprised of 86,000 noise panels. To do so, each of the noise panels undergoes the image conversion process, described in Section 2, producing six images each time that are then taken as

input to the predictive models. The summary statistics of the overall prediction results are presented in the form of histograms in Figure 5. The graphs on the left- and right-hand side show the probability distribution of the combined CNN predictions and the distribution of the body–wave panels detected above the probability threshold equal to 0.8, respectively.

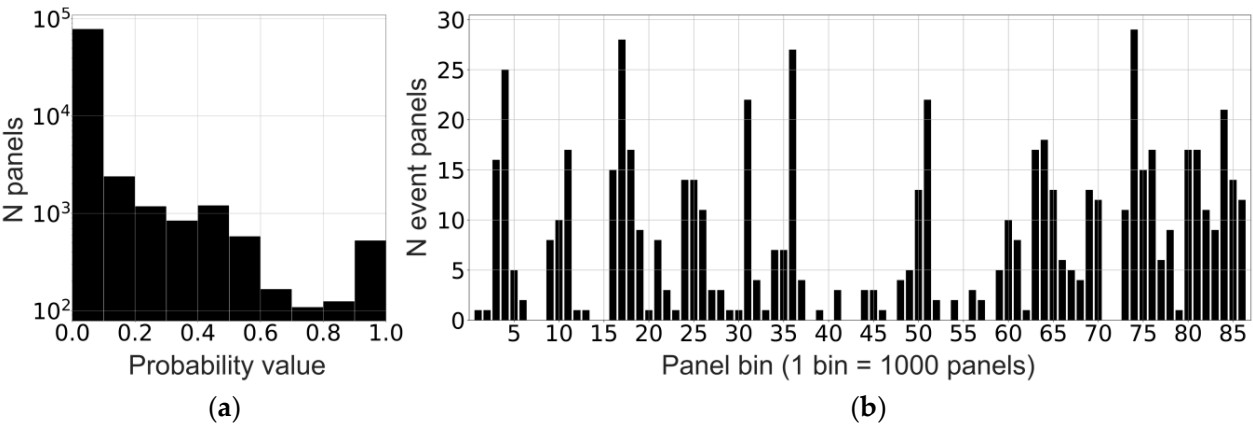

**Figure 5.** (**a**) Probability distribution of the body–wave occurrence in all analyzed noise panels provided by the ensemble of CNN-based models. (**b**) Distribution of the detected body–wave events (i.e., noise panels that were assigned with a probability estimate larger than 0.8).

### 4.3. Deep Embedded Clustering

In an attempt to group the detected body–wave panels according to their image characteristics, we perform clustering of the intermediate CNN outputs (activations) that are gathered for all predicted noise panels. These values (Figure 4, $AV_1$–$AV_6$) are extracted immediately after the first fully connected layer (Figure 4, $FCL_1$) as a half-product of the prediction routine. They give a view into how the input (i.e., attribute map) is decomposed into high-level features by building increasingly abstract representations with the subsequent network layers [64]. To facilitate the visualization and interpretation of clustering results, we first reduce the dimensionality of the 86,000 activation sets from 768 to 2 dimensions with the t-distributed stochastic neighbor embedding method [65] measuring pairwise local similarities between points in the high dimensional space. Although t-SNE projections are stochastic and influenced by the user-defined perplexity parameter, we choose the t-SNE specifically because of its widespread popularity and ability to preserve distinct neighborhoods in projections without overlapping [66]. In this way, we obtain a lower-dimensional t-SNE-embedded feature space, which can be easily visualized and used to infer the optimal number of k-clusters, which is a prerequisite for the employed k-means clustering algorithm [67]. As seen from Figure 6a, the projected samples of the detected and verified body–wave panel representations (depicted as yellow points) are accumulated at the bottom of the scatter plot and can be easily isolated from the other non-relevant samples (black points). Figure 6b provides a closer look at the yellow points that are assigned to the distinct groups obtained from the k-means cluster analysis with the predefined k equal to 5. We also compare the activation values extracted from training examples of attribute maps with the corresponding flattened attribute matrices before their conversion to images (Figure 1b, an example of an attribute matrix before flattening). In both cases, the batch of 1900 vectors of size 768 and 722, respectively, was mapped to 2-dimensional t-SNE space using the same settings (Figure 6c,d, respectively). On their basis, the two scatter plots are generated to visualize how the CNN rearranges the attribute representations of noise panels in the planar view.

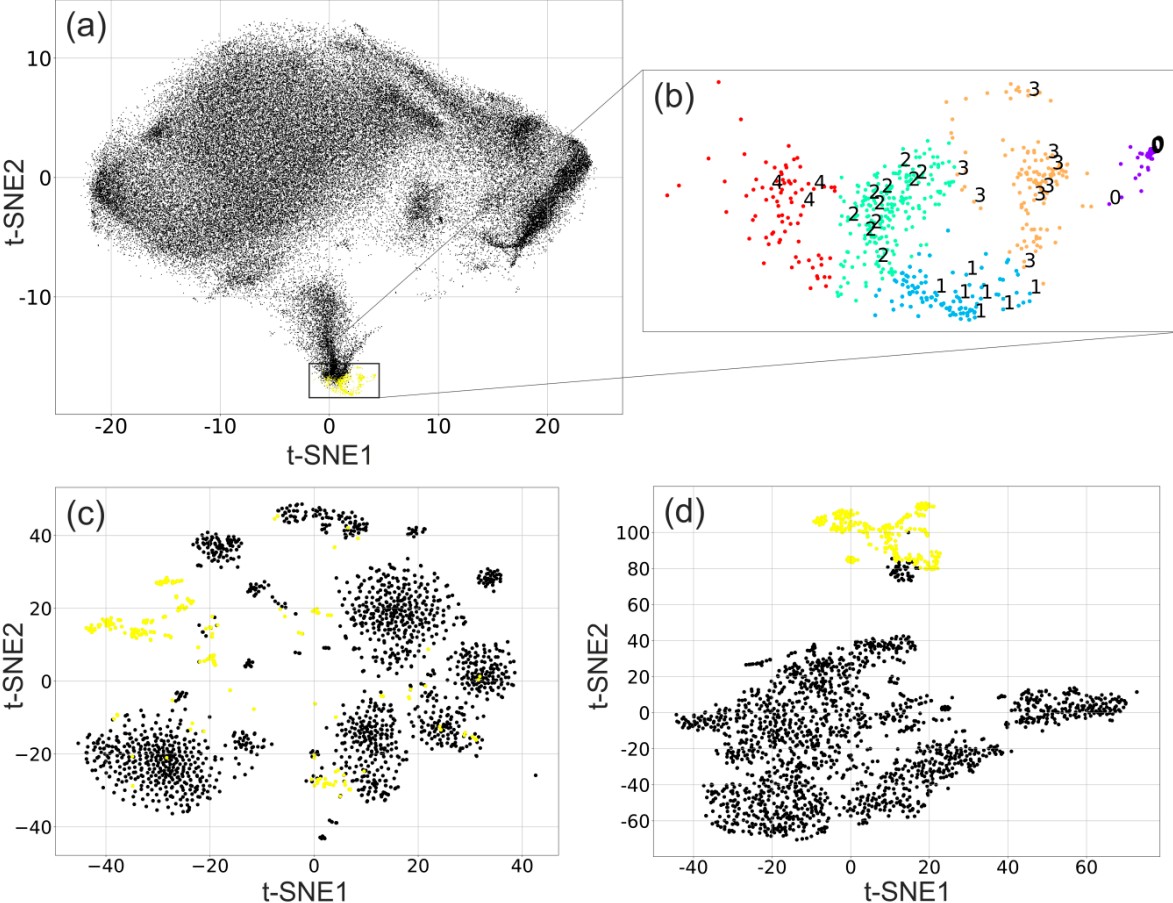

**Figure 6.** Visualization of the high-dimensional data using t-SNE. (**a**) t-SNE planar representation of the convolutional features extracted from 86,000 attribute maps (combined prediction half-products denoted as $AV_1$–$AV_6$ in Figure 5). The points plotted in yellow correspond to the noise panels classified as event panels. (**b**) K-means clustering of the detected body–wave panels. Each color represents a separate class. (**c**) t-SNE embedding of the initial feature matrix used for training the XGBoost models. The data matrix is comprised of 1900 flattened attribute maps that are not converted to images. (**d**) t-SNE embedding of the deep feature matrix extracted from 1900 images of attribute maps with the CNN-based models. The yellow points in (**c**,**d**) depict the same manually selected examples of body–wave panels.

To further investigate the prediction and clustering results, we present in Figure 7 the examples of the panels (unseen to the model during the training stage) classified as those in which the body–wave event happened. They are grouped into five image classes based on their spatial location in the t-SNE embedding (Figure 6b).

The visual inspection of the events divided into five clusters in Figure 7 suggest that predicted noise panels are grouped based on the two dominant image characteristics: (i) the SNR ratio of the body–wave events, and (ii) the presence of spurious arrivals that are represented by incoherent and coherent noise events. These spurious arrivals are events that are not associated with body–wave events and can be divided into three types: plane wave surface waves (cluster no. 1, and 2), airwaves (cluster no. 3), and high-amplitude bursts affecting up to several traces which are observed at whole time range (0–10 s) in detected body–wave panels (clusters no. 1–5). The last type of spurious arrival is the most frequent contribution observed in all panels, and thus it can be used as the "quality" indicator of the clusters, where the cluster no. 0 contains the smallest number of those arrivals, and for clusters with the increasing ordinal number, the contribution of these local noise events becomes more evident. This suggests that the proposed clustering methodology can be used as the provider of body–waves events required for SI reflection imaging, and as the data quality estimator.

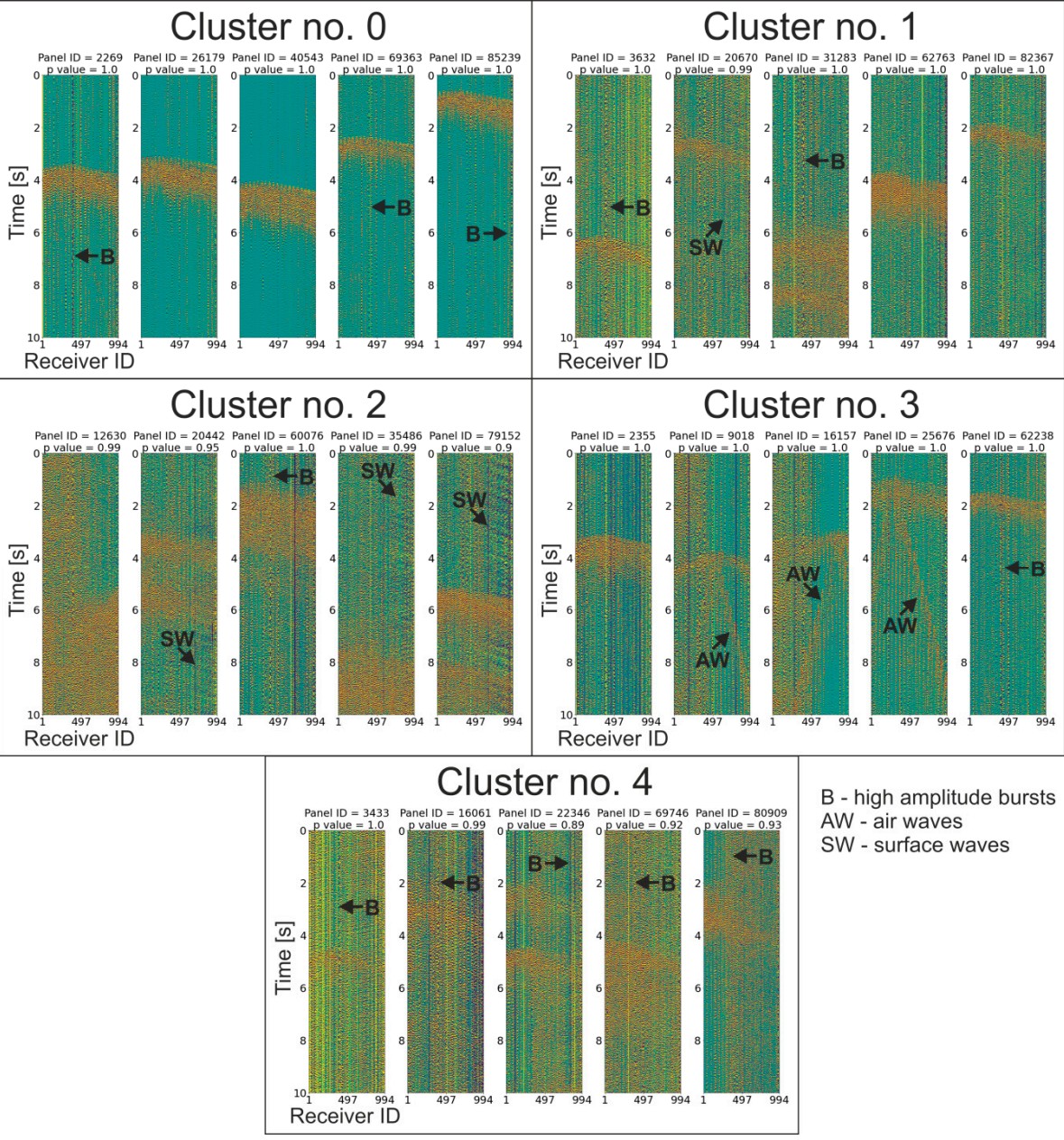

**Figure 7.** Examples of the body–wave events that are detected by the ensemble of CNN-based models. They are grouped into five image classes according to the clustering results shown in Figure 6b.

## 5. Discussion

Our study shows that CNNs trained on the hand-crafted attribute maps work well for the seismic event detection case. We generate these maps by exploiting typical seismic attributes like trace amplitude or frequency. Although we did not consider more sophisticated signal characteristics, we did investigate variations of the former ones obtained across various amplitude and frequency ranges or summarized with multiple descriptive statistics such as minimum, maximum, mean, median values, and standard deviation. To select the most representative combination of attributes, we trained several diagnostic CNN models that varied in terms of the number of the input heads (i.e., side branches of CNN architecture that are merged in the deeper layers) and we assessed them based on the learned high-level features. Given the fact that CNNs extract meaningful abstract features directly from the raw inputs and autonomously assign close to zero weights to the

irrelevant ones [68], we examined the activation values prior to the classification layer of the models at the two possible states, i.e., prediction of body–wave and non-body wave noise panels. We tried to retain those attributes that led to the activation values that allow for a clear distinction between the states and exclude those that generate mostly similar low-value activations irrespectively of an input. Based on the series of experimental tests, we found that the ensemble of amplitude- and frequency-based CNN models trained on the selected signal characteristics shown in Figure 2 provide the most optimal configuration.

In the present study, we analyzed the impact of the depth of CNN on performance scores by implementing the ResNet architecture. However, in the juxtaposition with the architecture presented in Section 3.2, the performance gain derived with the deeper one was not significant enough to accept the net of bigger complexity. To supplement the models' comparison, we also include the results of the prediction with the ensemble of XGBoost models, which achieved slightly worse scores on the validation data than the CNN-based counterpart (Table 3). The former detected over 21% less of the labelled body–wave panels. What is more, the exposure to the remaining data revealed much inferior generalizability of the XGBoost ensemble, probably due to the lack of ability to identify spatial patterns in the data.

In Figure 5a, we visualized the probability distribution of the CNN predictions for the entire unlabeled data set. Inferring from this summary we can notice that 1% of all noise panels (almost 1000 examples) were assigned with a probability value larger than 0.8, which we treat as the positively classified samples in this study. The conventional probabilistic boundary at 0.5 is still reasonable but it starts passing through the false positive (the non-event panels classified as the event ones). It is worth noticing that the training dataset was selected only at the interval between the first and the 40th-panel bin (Figure 5b), while the predicted events are more frequent in the second part, most likely related to the scheduled active-source shooting at the nearby mining site. It proves that the proposed ensemble of CNN-based models generalizes well to unseen data beyond the interval used for training.

Deep learning opens a new possibility of exploiting high-level features extracted at different layers of the designed net for clustering purposes as demonstrated in Section 4.3. With the dimensionality reduction method, we were able to visualize how the samples of the relevant class (i.e., body–wave panels) are reorganized when we compare the non-image input features (Figure 6c) with the features learned by the CNN-based models (Figure 6d). In terms of the practical implications, the clusters obtained in Figures 6b and 7 suggest that noise panels highlighted by t-SNE embedding represent the data useful for SI reflection imaging, however, due to their different level of contamination with other events, each cluster would require specific preprocessing approach before applying cross-correlation. For instance, due to the presence of airwaves, and surface waves, the preprocessing dedicated for clusters no. 1–3 will demand bandpass filtering in different frequency ranges and/or filtering-out different parts of f-k transform associated with varying moveouts of events that contaminate the body–wave energy. The removal of spurious, high amplitude bursts identified in Section 4.3, and appearing in each body–wave cluster shown in Figure 7 can be easily addressed with the trace energy normalization [1]. Nevertheless, the redundancy of the proposed solution evidenced by the similarity of the events within individual clusters, suggests that preprocessing of predicted panels can be automatized, and consequently our method can be used as the intermediate step towards developing the fully automatized ML-augmented ambient noise imaging workflow.

## 6. Conclusions

We have developed a deep learning technique for automatic data selection in massive ambient noise datasets recorded by a dense receiver array. It is designed to detect and isolate the time periods dominated by the body–wave energy based on the spatio-temporal characteristics of the signals that propagate across the seismic network. To provide these learnable patterns in the image form that is suitable for training a convolutional neural network, we preprocessed each noise panel in such a way that the constituent seismic

traces are represented by the six single-value attributes computed in the frequency and the time domain. The attribute values are then arranged in the corresponding attribute matrix and saved as color thumbnail images (i.e., attribute maps). During an exploratory data analysis, we selected almost 2000 training examples of the noise panels and we labeled them according to their class (i.e., body–wave or not-body–wave panel class). By addressing the problem of class imbalance with the oversampling method, we obtained the final training data set comprising almost 20,000 labeled images. The ensemble of two multi-headed CNN models trained separately on the frequency and amplitude attribute maps demonstrates better generalization ability than each of its participating networks. To fully benefit from the CNN learning process, we also showed that the high-level abstraction features, extracted at the deeper layers, can be combined and used to perform unsupervised clustering of the classified panels with respect to their image characteristics. We evaluated the model ensemble performance by retrieving the labels of noise panels that were held back from the training process using the 10-fold cross-validation technique. The proposed ensemble of CNN-based models achieved excellent F1 and AUC-PR scores of 0.963 and 0.997, respectively. It also outperformed the XGBoost ensemble that was built analogously but using models trained on non-image data (i.e., flattened attribute maps), and when applied to the full unlabeled dataset, it managed to find almost three times more verified body–wave events than it was provided at the training stage. The presented methodology can be used not only to optimize data selection approaches for long-term passive seismic studies but also allows an efficient real-time monitoring technique for revealing spatio-temporal variations in multiple scattering media to be established. It may be especially valuable for a range of applications, where the robust retrieval of body waves is crucial for estimating trustful 3-D velocity structure and tracking its changes. The CNN-based solution for body–wave detection in ambient noise recordings proved its ability to identify those time segments in continuous data that contain the most usable energy for the reconstruction of the seismic Green's functions.

**Supplementary Materials:** The paper provides the database used in the current study and the Jupyter Notebook python code for training the CNN models available online at GitHub (https://github.com/mmezyk/ansi_selector).

**Author Contributions:** Conceptualization, M.M. (Miłosz Mężyk) and M.C.; methodology, M.M. (Miłosz Mężyk) and M.C.; software, M.M. (Miłosz Mężyk); validation, M.M. (Miłosz Mężyk) and M.C.; formal analysis, M.M. (Miłosz Mężyk); investigation, M.M. (Miłosz Mężyk); resources, M.M. (Michał Malinowski); data curation M.M. (Miłosz Mężyk) and M.C.; writing—original draft preparation, M.M. (Miłosz Mężyk), M.C., M.M. (Michał Malinowski); writing—review and editing, M.M. (Miłosz Mężyk), M.C., M.M. (Michał Malinowski); visualization, M.M. (Miłosz Mężyk); supervision, M.M. (Michał Malinowski); project administration, M.M. (Michał Malinowski) All authors have read and agreed to the published version of the manuscript.

**Funding:** This work was supported by the Polish National Science Centre (NCN) under grant no. UMO-2018/30/Q/ST10/00680. Cost of publication was covered by the Polish National Agency for Academic Exchange (NAWA) under grant no. PPI/PZA/2019/1/00107/U/00001.

**Institutional Review Board Statement:** Not applicable.

**Informed Consent Statement:** Not applicable.

**Data Availability Statement:** The data are confidential and not shareable due to privacy concerns.

**Acknowledgments:** Kylylahti large-N array was a part of the COGITO-MIN project, which has been funded under the ERA-MIN network and received funding from TEKES (Business Finland) (Finland) and NCBR (Poland).

**Conflicts of Interest:** The authors declare no conflict of interest. The funders had no role in the design of the study; in the collection, analyses, or interpretation of data; in the writing of the manuscript, or in the decision to publish the results.

**Abbreviations**

| | |
|---|---|
| AM | Attribute Map |
| ANSI | Ambient Noise Seismic Interferometry |
| ATV | Attribute Value |
| AUC-PR | Area Under Curve—Precision-Recall |
| AV | Activation Value |
| AW | Air Wave |
| CCF | Cross-Correlation Functions |
| CNN | Convolutional Neural Network |
| CL | Convolution Layer |
| CPU | Central Processing Unit |
| CV | Cross-Validation |
| DAS | Distributed Acoustic Sensing |
| DL | Dropout Layer |
| EGF | Empirical Green's Function |
| FCL | Fully Connected Layer |
| FN | False Negative |
| FP | False Positive |
| MEMS | Micro-Electromechanical System |
| ML | Machine Learning |
| MPL | MaxPooling Layer |
| PR | Precision-Recall |
| RAF | ReLU Activation Function |
| ReLU | Rectified Linear Unit |
| RGB | Red-Green-Black |
| RMS | Root-Mean-Square |
| ROC | Receiver Operator Characteristic |
| SAF | Softmax Activation Function |
| SGD | Stochastic Gradient Descent |
| SI | Seismic Interferometry |
| SNR | Signal-to-Noise Ratio |
| SVM | Support Vector Machine |
| SW | Surface Wave |
| TN | True Negative |
| TP | True Positive |
| t-SNE | t-Distributed Stochastic Neighbor Embedding |
| VGG | Visual Geometry Group |
| XGBoost | eXtreme Gradient Boosted Tree |

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
