# Peer review of "Automatic Image-Based Event Detection for Large-N Seismic Arrays Using a Convolutional Neural Network"

_remotesensing, doi:10.3390/rs13030389_

Round 1
Reviewer 1 Report
Thank you for contributing such a great paper to Remote Sensing. This work is outstanding and well-written. I have a few comments and hopefully be useful to improve the paper more or less.
First of all, I want to clarify the body wave events used in the paper are the real mining blasts based on my reading, instead of SI’s body wave data. This could be misleading in the current writings. I am fine with the introduction to motivate the importance of identification of body wave. But somewhere authors should claim what’s the input data to be used here. It’s not a bad idea to discuss the applicability of this proposed methodology to the SI body-waves in the Discussion section.
“utilizing the map-view projection for the equirectangular image representation of noise panels”.
I am not sure what this is. Does this mean each trace will be processed individually and then combine the attribute of each receiver into the map? Please clarify this. This also need to explain the Figure 3a &b. How does this event data illustrate this procedure?
Line 150: “apply transformations”, what’s the transformations?
Line 156, why 722 receivers from 994?
If authors prefer to stay with gray-black, Figure 2’s gray curve could be lighter to increase the contrast.
Frequency range should be 10-20, 20-30, 30-40 Hz. Please double check the text with an inconsistent freq range.
Figure 2f: x label should be 0, 0.25, 0.75, 1.00.
It is not very clear to me how (g) leads to 3 attributes and (f) lead to 2 attributes. They (f) and (g) seems to have a similar data structure.
Line 320, missing a period ‘.’.
What does Figure 5 tell us? I didn’t follow the 0.8 very well. I am wondering whether we identify most of our known events or more than what we knew?
I love Figure 6. What’s the 5 group from k cluster? Any implication of events? Why 5? What’s the difference between (c) and (d)?
Figure 7 is very informative. Could you use arrows to point out the waves (surface wave, air, burst, spurious arrivals etc.) in the figure?
After I am reading the manuscript, I recalled a relevant paper to identify the P-wave arrivals by classify the signal and noise using CNN. They share the similarity and I hope authors find it be relevant.
Guo C., Zhu T., Gao Y., Wu S., and Sun J., AEnet: Automatic picking of P-wave first arrivals using deep learning, IEEE Geoscience and Remote Sensing, 10.1109/TGRS.2020.3010541
Reviewer 2 Report
The manuscript entitled “Automatic image-based event detection for large-N seismic ar-2 rays using a convolutional neural network” investigated a CNN-based model for damage event detection under natural hazards.
After careful review, the manuscript has a reasonable effort and technical information. Firstly, I am not sure by having such a high accuracy just by CNN and fundamental optimization about the accuracy of the data and applicability of the method. It has to be clarified in the next response letter. However, there is not much novelty on the work, and there are some points that must be considered in the revision to be worth being accepted. Therefore, I strongly recommend the authors to follow the comments below:
1- As you have many abbreviations, therefore please provide an abbreviation table that on MDPI format must be at the end of the paper, however you made it but is not complete.
2- Abstract: The sentence is a bit confusing and is very long. You can concise it and point the important parts.
3- The work presents a poor literature review on the methods based on soft computing (ML, Fuzzy and AI) for damage assessment. Because your work is not much new and novel, I strongly recommend you make your introduction and study on previous methods interesting for readers by adding the following new works which I found it new and related to your work which are based on different methods for rapid damage assessment of buildings. It will increase the dept of your review and stronger and wider area.
-Multi-Hazard and Spatial Transferability of a CNN for Automated Building Damage Assessment
-Application of Support Vector Machine Modeling for the Rapid Seismic Hazard Safety Evaluation of Existing Buildings
-Harirchian, E.; Kumari, V.; Jadhav, K.; Raj Das, R.; Rasulzade, S.; Lahmer, T. A Machine Learning Framework for Assessing Seismic Hazard Safety of Reinforced Concrete Buildings. Appl. Sci. 2020, 10, 7153.
-Bai, Y.; Hu, J.; Su, J.; Liu, X.; Liu, H.; He, X.; Meng, S.; Mas, E.; Koshimura, S. Pyramid Pooling Module-Based Semi-Siamese Network: A Benchmark Model for Assessing Building Damage from xBD Satellite Imagery Datasets. Remote Sens. 2020, 12, 4055.
-Nex, F., Duarte, D., Tonolo, F. G., & Kerle, N. (2019). Structural building damage detection with deep learning: Assessment of a state-of-the-art cnn in operational conditions. Remote sensing, 11(23), 2765.
-CABiNet: Efficient Context Aggregation Network for Low-Latency Semantic Segmentation
-Shin, H. C., Roth, H. R., Gao, M., Lu, L., Xu, Z., Nogues, I., ... & Summers, R. M. (2016). Deep convolutional neural networks for computer-aided detection: CNN architectures, dataset characteristics and transfer learning. IEEE transactions on medical imaging, 35(5), 1285-1298.
-Valentijn, T. (2020). The Practical Applicability of a CNN for Automated Building Damage Assessment.
4- Figure 2 and 3: Please improve the quality and font size to be more visible
5- Please provide a figure that shows the architecture of your model and network. However figure 4 is presenting such a thing but needs more description and make it more understandable.
6- Please make your tables and figures follow a same path and font size and colors
7- There are many typos that needs to be corrected e.g. line 459 you write Figure 6 c, but in line 469 is written Fig.7, please make the format similar for all. Also, be careful about spaces after (.) amd (,) which in some cases are doubled or missed.
8- Generally, it seems that you have a very high accuracy and it is a bit strange because you did not present the ROC of each class and did not control the overfitting. Especially in the case of using different data for training and test! Please clarify more in this manner.
9- You did not highlight the problem statement, objectives and novelty of your proposed method.
10- There is a need in proofreading the work.
11- Please provide more information about the data collection or case study if they are from some specific region. However, you have presented in
Figure 1 but needs more clarifications and I recommend to sumarize the lable of figure and write more in the body of the text.
12- Please provide more information about the analysis rather than just F1 score. Show your confusion matrix and the parameters you can achieve from it.
At the end as I have mentioned, there are not much significant novelty on this work but the efforts were good and it would be good if you revise it according to the points provided and other reviewers.
Reviewer 3 Report
The study generates an interesting argument. Comparative with other methods used in AI could have been interesting to see how the fair. All I see is a clustering method without comparison to other methods. How do the justify that this method is better than others.
Round 2
Reviewer 2 Report
Dear Authors
Many thanks for your responses and efforts you made to brings up your paper into an acceptable level. Please write down the abbreviation table in the original paper for publishing purposes according to MDPI style. About the recommended papers, as you were agree with me, your literature due to the lack of available similar works are less, therefore you can cover some of the related works which used machine learning and CNN techniques to show up the broad application of these methods. However, we can understand that you might concern about your field but according to my experience, using such recommended works will help for your paper to get the attention of other readers and make your work highly citable.
About the font size of figures we can understand your response but if you look on it the size of (a) or (b) in figures are like 18-20 and is a bit annoying in compare to the rest of texts. You can make it 14 to be visible and has less contradictions.
The rest I hope you follow my comments and make your paper for publishing properly.
Author Response
Dear Reviewer,
after all, we decided to include additional 10 references in the introduction. They are related to the application of deep learning to earthquake detection problems that we find more compliant with our work than what was already proposed (i.e. damage detection issues). We put them in one group to emphasize their importance and contribution to AI research.
With regard to the two other remarks, we normalized the font size of the three figures and we included the abbreviation table in the manuscript just after the conclusions. We hope it is in accordance with the MDPI style now.
On behalf of the authors,
Miłosz Mężyk